# Evaluating Diagnostic Ultrasound of the Vagus Nerve as a Surrogate Marker for Autonomic Neuropathy in Diabetic Patients

**DOI:** 10.3390/medicina59030525

**Published:** 2023-03-08

**Authors:** Bianka Heiling, Adriana Karl, Nadin Fedtke, Nicolle Müller, Christof Kloos, Alexander Grimm, Hubertus Axer

**Affiliations:** 1Department of Neurology, Jena University Hospital, Friedrich Schiller University, 07747 Jena, Germany; 2Clinician Scientist Program OrganAge, Jena University Hospital, 07747 Jena, Germany; 3Department of Internal Medicine III, Jena University Hospital, Friedrich Schiller University, 07747 Jena, Germany; 4Department of Neurology, Tübingen University Hospital, 72076 Tübingen, Germany

**Keywords:** peripheral nerve ultrasound, vagus nerve, diabetes mellitus, autonomic neuropathy

## Abstract

*Background and Objectives*: Diagnostic ultrasound of the vagus nerve has been used to examine different polyneuropathies, and it has been suggested to be useful as a marker of autonomic dysfunction in diabetic patients. *Materials and Methods*: We analyzed the cross-sectional area (CSA) of the right vagus nerve of 111 patients with type 2 diabetes in comparison to 104 healthy adults and 41 patients with CIDP (chronic inflammatory demyelinating polyneuropathy). In the diabetes group, sympathetic skin response (SSR) was measured as an indicator for autonomic neuropathy. Carotid intima–media thickness (CIMT) was measured as a surrogate for atherosclerosis. Clinical symptoms of polyneuropathy were assessed using the Neuropathy Symptom Score and the Neuropathy Disability Score. *Results*: In total, 61.3% of the diabetes patients had clinical signs of polyneuropathy; 23.4% had no SSR at the feet as an indicator of autonomic neuropathy. Mean vagus nerve CSA did not differ in patients with and without diabetic polyneuropathy or in diabetic patients with and without SSR at the feet. No significant correlation was found between vagus nerve CSA and CIMT or SSR parameters in diabetic patients. Mean CSA of the right vagus nerve was slightly larger in diabetic patients (*p* = 0.028) and in patients with CIDP (*p* = 0.015) than in healthy controls. *Conclusions*: Effect sizes and mean differences were rather small so that a reliable diagnosis cannot be performed based on the vagus nerve measurement of a single person alone. Vagus nerve CSA seems not suitable as an indicator of autonomic dysfunction or cardiovascular risk in diabetic patients.

## 1. Introduction

The prevalence of diabetes mellitus and its complications is increasing with a growing of the public health burden of diabetes across the world. The prevalence of diabetes worldwide was estimated to be 2.8% in 2000, i.e., 171 million people, and will presumably be increasing to 4.4% in 2030, i.e., 366 million people [1].

A major complication of diabetes is diabetic neuropathy. It has been estimated that approximately 350 million patients will develop diabetic neuropathy until 2045 [2]. In patients with newly diagnosed type 2 diabetes, the prevalence for diabetic neuropathy is approximately 20% and prevalence increases up to 50% after 10 years or more [2].

In addition, the impairment of the autonomic nervous system in the course of diabetic neuropathy is detrimental because cardiovascular autonomic neuropathy leads to increased morbidity and mortality based on a high risk for arrhythmia and sudden cardiac death [3]. A decreased heart rate variability is an early indicator. Cardiovascular autonomic neuropathy may be found in approximately 20% of patients with diabetes mellitus and its prevalence increases with age and duration of diabetes [4].

The autonomic nervous system innervates most of the inner organs mainly by efferent autonomic nerve fibers from both the sympathetic and parasympathetic nervous system. Dysfunction of efferent autonomic innervation leads to trophic changes (such as diabetic foot syndrome, edema, ulcer, osteoarthropathy), cardiovascular dysregulation (tachycardia at rest, decreased heart rate variability, and others), gastrointestinal problems (esophageal dystonia, gastroparesis, diarrhea, constipation), and urogenital symptoms (urinary incontinence, erectile dysfunction) [3,4]. Impairment of afferent autonomic nerve fibers may cause missing pain in myocardial infarction, missing reactions in hypoglycemia, or the loss of bladder control. Thus, symptoms of autonomic dysregulation in diabetes mellitus may be manifold and often are not reported spontaneously by patients. As many of these dysfunctions are potentially harmful, an easy diagnostic tool for detection of autonomic dysregulation would be very useful. Recently, it was reported that the mean cross-sectional area (CSA) of the vagus nerve measured with ultrasound was significantly smaller in a study of 54 patients with diabetes compared to 20 healthy controls [5]. The authors attributed this finding to an atrophic change of the vagus nerve in diabetic patients. Similar findings were also described for transthyretin familial amyloid polyneuropathy [6].

We, therefore, intended to evaluate if this finding could be reproduced in a larger cohort of patients with type 2 diabetes and if the vagus nerve CSA may be associated to diabetic neuropathy, cardiovascular events such as coronary heart disease and stroke, the existence of autonomic neuropathy measured by means of sympathetic skin response (SSR), and other clinical parameters.

## 2. Materials and Methods

### 2.1. Patients

Patients with type 2 diabetes who presented in the outpatient clinic for diabetology at Jena University Hospital between September 2020 and December 2022 were included in the study. Patients between 40 and 85 years of age were included, who were willing to fill out questionnaires and to undergo electrodiagnostic testing and ultrasound examination. Other known etiologies for polyneuropathy (such as alcohol abuse, inflammatory polyneuropathies, etc.), rheumatic disease, peripheral arterial occlusive disease, active malignant tumor disease, and history of chemotherapy were exclusion criteria. The data were prospectively collected (SELECT study [7], German Clinical Trials Register DRKS00023026). All participants gave written informed consent. The study was approved by the local ethics committee (number 2019-1416-BO).

### 2.2. Assessments

Age, gender, duration of diabetes in years, body mass index (kg/m^2^), HbA_1c_ (mmol/mol), and glomerular filtration rate (ml/min) were collected. In addition, participants were asked for their history of coronary heart disease and stroke.

The Neuropathy Symptom Score (NSS) was used to quantify the symptoms due to diabetic polyneuropathy [8]. The NSS recognizes sensory symptoms in the legs (burning, numbness, tingling, fatigue, cramping), the localization, the time of appearance, and improvements. The Neuropathy Disability Score (NDS) quantifies the severity of sensory deficits (ankle reflexes, vibration perception threshold (tuning fork), pain sensitivity (pin-prick), and temperature sensitivity) [8]. If the NDS is between 6 and 8 or the NDS is between 3 and 5 and the NSS is between 5 and 6, diabetic polyneuropathy is diagnosed from a clinical point of view [9].

### 2.3. Ultrasound

A high-resolution ultrasound device (Mindray M7, Ultrasound systems, Darmstadt, Germany) with a 14 MHz linear-array transducer was used for sonographic examinations of the patients with diabetes. The normal values of the vagus nerve were analyzed with a 14 MHz probe with a TE7 (Mindray T7, Ultrasound systems, Darmstadt, Germany). Patients with chronic inflammatory demyelinating polyneuropathy (CIDP) were examined using an 18 MHz high-resolution probe (Canon Aplio 800, Canon Medical Systems USA Inc., Tustin, CA, USA). All examinations were conducted by an experienced neurologist.

The cross-sectional area (CSA) of the right vagus nerve (Figure 1A) was measured as follows: Patients were scanned in the supine position and were asked to turn their head to the left side. To record the nerve in the axial view, the probe was first placed on the midcervical plane (lateral to the thyroid cartilage). Landmarks for identifying the nerve were the carotid artery and the internal jugular vein. Here, the vagus nerve lies within the carotid sheath between the artery and vein, visible as a small, round hypoechoic structure, sometimes having a honeycomb structure. The settings of the ultrasound device were optimized individually for each patient in terms of gain, depth, and focus. Vagus nerve CSA was determined by tracing the nerve area within the hyperechoic epineurium at an axial section.

In addition, the carotid intima–media thickness (CIMT) of the right common carotid artery (ACC, Figure 1B) was measured as an indicator for atherosclerosis. For this purpose, the ACC was approached axially at midneck level. Next, the probe was rotated 90° to display the ACC longitudinally and placed vertically approximately 2 cm below the bifurcation. The CIMT was then measured at the wall far from the transducer.

### 2.4. Sympathetic Skin Response and Nerve Conduction Studies

The sympathetic skin response (SSR, Figure 1C,D) [10] at the right hand and the right foot was measured as a parameter for autonomic neuropathy. The measurements were performed by an experienced neurologist using a Medelec Synergy device (Synergy 15.0; Viasys Healthcare, Natus Europe GmbH, Planegg, Germany).

Surface electrodes were placed on the patients’ right palm and the sole of the foot with a reference electrode at the dorsum of the hand or the foot, respectively [11,12]. Sympathetic stimulation was applied at the wrist by a 30 mA single-pulse electrical stimulation, 0.1 ms in duration. Four stimulations were delivered randomly and at a minimal interstimulus interval of more than 30 s. Room temperature was between 22 and 24 °C. Amplitude and latency of the most representative measurement were collected.

In addition, nerve conduction studies (NCS) were performed. Motor and sensory nerve responses were assessed for the median nerve, motor responses for the tibial nerve, and sensory nerve responses for the sural nerve. NCS positive neuropathy was diagnosed if at least two nerves showed pathological measurements (according to the reference values in our neurophysiology lab).

### 2.5. Controls

Age, gender, and vagus nerve CSA were taken from an existing database of 104 healthy controls, which have been already described in detail [13]. Data from healthy subjects (University Hospital Tübingen) consisting of medical staff and patients without neuromuscular disorders have been collected in this database.

In addition, we retrospectively collected age, gender, and vagus nerve CSA from 41 patients (treated at Jena University Hospital) suffering from CIDP (chronic inflammatory demyelinating polyneuropathy) as a control group with inflammatory polyneuropathy. CIDP was confirmed in each patient based on the EFNS criteria [14]. This retrospective analysis of patient data has been approved by the local ethics committee (number 2022-2790-Daten). 

### 2.6. Statistics

The Statistical Package for the Social Sciences software (SPSS version 25.0; IBM Corporation, Armonk, NY, USA) was used for all analyses. Values are presented as mean and standard deviation (SD) and numbers and percentages. Baseline parameters of the cohort were quantified using descriptive statistics. Pearson correlations were used to analyze correlations between vagus nerve CSA, SSR parameters, CIMT, and other clinical parameters. Unpaired *t*-test was used to analyze differences in vagus nerve CSA between groups. For all analyses, a *p* value < 0.05 was considered statistically significant. Cohen’s d was used as an estimate of the effect size. 

## 3. Results

### 3.1. Patients

A total of 111 patients with type 2 diabetes mellitus were enrolled in the study. Table 1 shows the descriptive statistics of the parameters. Among patients, 61.3% had clinical signs of polyneuropathy, 18.1% suffered from coronary heart disease, 10.8% had a medical history of stroke. In 26 patients (23.4%), no SSR could be detected at the feet as an indicator of autonomic neuropathy.

### 3.2. Interactions

We did not find (Figure 2) any statistically significant difference of vagus nerve CSA in diabetic patients with and without clinical signs of polyneuropathy (*p* = 0.158), in diabetic patients with and without coronary heart disease or stroke (*p* = 0.073), and in diabetic patients with and without SSR at the feet (*p* = 0.850). Appendix A shows these data separated according to gender. Even in patients with and without NCS-based polyneuropathy, a significant difference of vagus nerve CSA could not be found (*p* = 0.146). Details of the relationship between clinically diagnosed and NCS-based-diagnosed polyneuropathy can be seen in Appendix A. 

In addition, there was no correlation between vagus nerve CSA and any of the analyzed parameters (Table 2). The scatterplots of vagus nerve CSA and all the parameters analyzed are shown in Appendix A. A significant correlation was found between age and duration of diabetes, HbA_1c_, glomerular filtration rate, and CIMT. These scatterplots are shown in Appendix A. The amplitude of SSR at the foot was correlated with the amplitude of the SSR at the hand.

In addition, a regression analysis using vagus nerve CSA as a dependent variable, and SSR parameters and CIMT as independent variables was performed (see Appendix A). No significant beta-coefficient was found.

### 3.3. Controls

Table 3 shows the baseline characteristics of the 111 patients with diabetes mellitus, the 104 healthy controls, and the 41 CIDP patients. 

Mean vagus nerve CSA was slightly larger in the diabetes patients compared to the healthy controls (*p* = 0.028) with a small effect size (Cohen’s d 0.302). Mean vagus nerve CSA was larger in the CIDP group compared to the healthy controls (*p* = 0.015) with a medium effect size (Cohen’s d −0.456). Figure 3 shows the box plots of vagus nerve CSA in the three groups. Appendix A shows these data separated according to gender. Vagus nerve CSA showed a statistically significant sex-dependent difference for the normal controls (*p* = 0.022), but neither for the diabetes nor for the CIDP patients.

## 4. Discussion

Diagnostic ultrasound of the vagus nerve has been applied in polyneuropathies of different etiologies such as Guillain–Barré Syndrome (GBS) [15], CIDP [16], transthyretin familial amyloid polyneuropathy [6], and also in patients with Parkinson’s disease [17,18,19]. It was suggested that vagus nerve enlargement may be a useful marker for autonomic dysfunction in patients with GBS [15,20].

The vagus nerve represents the main parasympathetic division of the autonomic nervous system. It contains predominantly (80–90%) afferent sensory fibers and, to a lower degree, (10–20%) efferent fibers [21]. The cervical vagus nerve innervates heart, tracheobronchial tree and lungs, liver, pancreas, and the gastrointestinal tract up to the proximal colon. 

Therefore, the hypothesis that vagus nerve CSA is altered as a marker for autonomic impairment in patients with diabetes mellitus seems reasonable. In our study, 23.4% of the diabetic patients had a loss of SSR at the feet, indicating autonomic neuropathy, which is in line with the literature reporting a prevalence of approximately 20% of cardiovascular autonomic neuropathy in patients with diabetes mellitus [4].

Tawfik et al. [5] found the mean CSA of the right vagus nerve to be smaller in 53 patients with diabetes mellitus (2.1 mm^2^ ± 1.1) compared to 20 healthy controls (5.8 mm^2^ ± 1.3). In our study, mean vagus nerve CSA was 2.49 mm^2^ ± 0.88 in 111 patients with diabetes mellitus, which is slightly larger. However, the mean vagus nerve CSA of 104 healthy controls (2.26 mm^2^ ± 0.59) was considerably smaller in our study than the CSA reported in the study of Tawfik et al. [5]. Thus, we could not reproduce the described large difference of vagus nerve CSA between diabetic patients and normal controls in a much larger cohort. In particular, we were not able to find atrophic changes in the vagus nerve of diabetic patients.

A recent meta-analysis estimated the overall mean CSA of the right vagus nerve of healthy subjects as 2.53 mm^2^ (with a 95% confidence interval of 2.11–2.95 mm^2^) at the level of bifurcation of the carotid artery and as 2.39 mm^2^ (with a 95% confidence interval of 2.24–2.55 mm^2^) at the level of the thyroid gland [22]. Another meta-analysis calculated the mean pooled CSA of the vagus nerve as 2.2 mm^2^ (with a 95% confidence interval of 1.5–2.9 mm^2^) [23]. All these estimates of vagus nerve CSA are much smaller than the CSA measurements in the control group of Tawfik et al. [5], which may possibly be due to the relatively small number of 20 healthy controls in that study. In addition, it has been noted that a great variation for the vagus nerve CSA is reported in the literature, and factors such as age, body side, ultrasound settings, and imaging technique used may impact the accuracy of CSA measurement [24]. 

In our study, mean CSA of the right vagus nerve was larger in diabetes patients and in patients with CIDP than in the healthy controls. However, effect size and mean differences were rather small, so that no reliable diagnosis can be drawn from the vagus nerve measurement of a single person alone. The largest mean vagus nerve CSA was found in the patients with CIDP, which is to be expected in a patient group with inflammatory polyneuropathy [25].

Examinations of the cardiovascular autonomic system comprise measurements of heart rate variability (HRV) at rest, during deep metronomic breathing (calculation of the quotient between the longest heart rate interval during inhalation and the shortest interval during exhalation, I:E Ratio), during orthostatic maneuver (calculation of the quotient of the longest heart rate interval at 30 heartbeats and the shortest heart rate interval at 15 heartbeats, 30:15 ratio), and during Valsalva maneuver (calculation of the quotient of the longest heart rate interval after the maneuver and the shortest heart rate interval during the maneuver) [26]. 

However, these HRV examinations are complex and time-consuming and the SELECT study was primarily not conceived to assess cardiovascular autonomic dysfunction. Measurements of SSR and CIMT are relatively easy to perform in the study of using ultrasound as a diagnostic tool for diabetic polyneuropathy and are described as surrogate markers for autonomic impairment and atherosclerosis.

The sympathetic skin response (SSR) measures electrodermal activity evoked by sympathetic stimulation such as electrical stimulation [26]. It has been shown recently that SSR may be used as an indicator for diabetic cardiac autonomic neuropathy [27,28].

Carotid intima–media thickness (CIMT) is a widely used marker of atherosclerosis [29,30]. Diabetic polyneuropathy was found to be associated to CIMT in patients with type 2 diabetes [31,32,33]. In addition, it was described that cardiac autonomic neuropathy in patients with type 2 diabetes is also associated with carotid atherosclerosis, represented as CIMT [34,35].

In this study, we found no correlations of the right vagus nerve CSA either to CIMT nor to any of the SSR parameters in the diabetic patients. In addition, a regression analysis using vagus nerve CSA as a dependent variable, and all of these parameters collected as independent variables showed no significant beta-coefficients, which also confirms that there may be no clinical useful relationship between these parameters and vagus nerve CSA.

In addition, vagus nerve CSA was not different in diabetic patients with and without polyneuropathy, with and without coronary heart disease or stroke, and with and without SSR at the feet. Overall, we did not find any proof that vagus nerve CSA may be suited to serve as a robust indicator for any of the measured parameters related to autonomic dysfunction or cardiovascular risk.

Some limitations of the study have to be discussed. It has to be kept in mind that the methods used in this study investigate a limited part of autonomic dysregulation only. Other elements of autonomic neuropathy in diabetics, such as gastrointestinal, genitourinary and ocular, the affection of thermoregulation and blood pressure, constipation, and sexual dysfunction have not been analyzed. 

The SELECT study was not designed with a special focus on vagus nerve ultrasound and autonomic dysregulation. Therefore, the right vagus nerve only was analyzed. It has been reported that the right vagus nerve CSA is larger than the left measured by means of ultrasound [22,36,37]. Interestingly, such a side difference could not necessarily be reproduced in cadaveric studies [38]. In contrast, vagus nerve branching was found to have an influence on the cross-sections of the vagus nerve, with the cross-sections with branching being considerably larger [38]. 

In addition, data on diabetic patients, healthy controls, and CIDP patients were not collected in the same study and different ultrasound probes were used, which may introduce some systematical bias. Nevertheless, results of CSA measurements did not extensively differ from other larger studies discussed above, and good intrarater, interrater, and across-ultrasound systems agreements have been shown earlier [37].

## 5. Conclusions

Overall, the finding of vagus nerve atrophy in diabetic patients was not confirmed in this study. In addition, we did not find that vagus nerve CSA may be suited as an indicator for any of the measured parameters indicating autonomic dysfunction or cardiovascular risk. Thus, vagus nerve CSA alone may not be suited as a marker for autonomic dysfunction in diabetic patients.

## Figures and Tables

**Figure 1 medicina-59-00525-f001:**
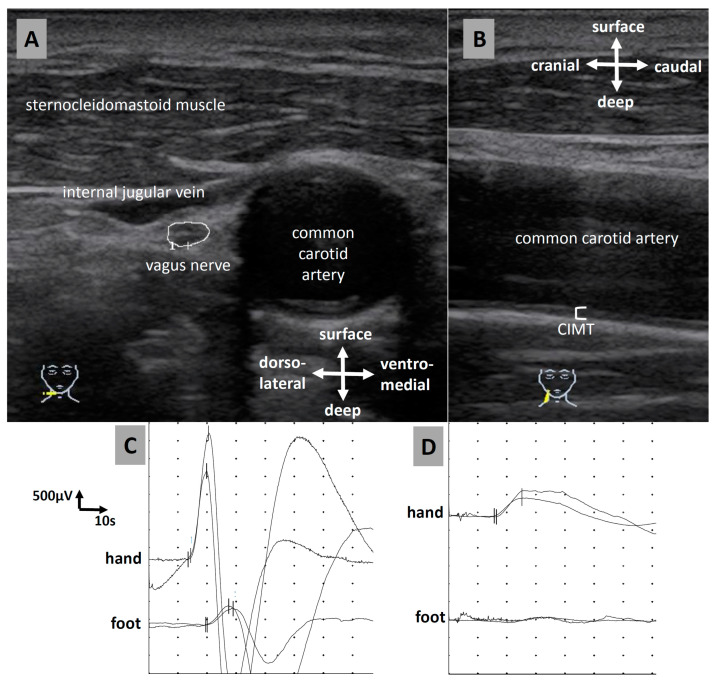
Ultrasound measurements and sympathetic skin response (SSR). (**A**) Ultrasound of the vagus nerve. (**B**) Carotid intima–media thickness (CIMT). (**C**) Normal SSR. (**D**) Pathological SSR with missing responses at the feet.

**Figure 2 medicina-59-00525-f002:**
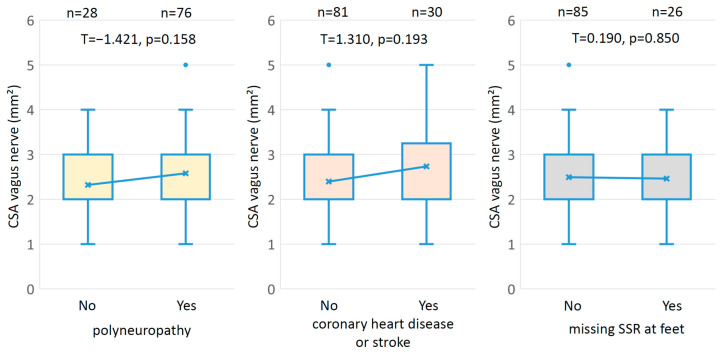
Box plots of vagus nerve CSA in different subgroups of patients with type 2 diabetes mellitus. Note that there were no statistically significant differences of vagus nerve CSA in all of these conditions.

**Figure 3 medicina-59-00525-f003:**
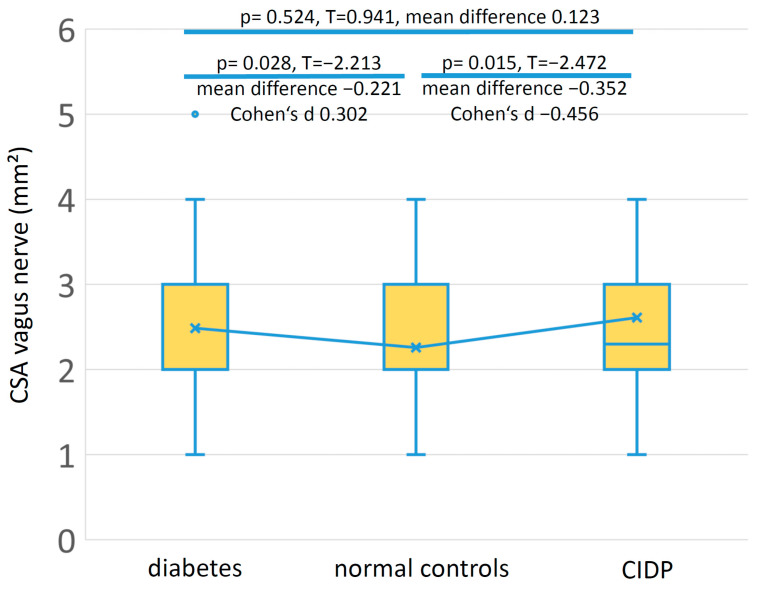
Box plots of vagus nerve cross-sectional area (CSA) in patients with type 2 diabetes, normal controls, and patients with chronic inflammatory demyelinating polyneuropathy (CIDP). Although the differences of vagus nerve CSA between diabetes patients and normal controls as well as between CIDP patients and normal controls were statistically significant, the effect sizes were rather small.

**Table 1 medicina-59-00525-t001:** Baseline characteristics of the patients with type 2 diabetes mellitus (*n* = 111).

**Categorical Variables**				**N**	**%**
Gender	Female/Male			43/68	38.7/61.3
Polyneuropathy if NDS > 5 or (NDS > 2 and NSS > 4)	Yes/No			76/28	61.3/38.7
NCS-based diagnosis of polyneuropathy	Yes/No			80/31	72.1/27.9
Known coronary heart disease	Yes/No			21/90	18.1/81.9
Stroke in history	Yes/No			12/99	10.8/89.2
**Metric Variables**	**Mean**	**SD**	**Minimum**		**Maximum**
Age (years)	66.31	10.05	42		84
Duration of diabetes (years)	16.08	9.46	0.5		39.6
Body mass index (kg/m^2^)	32.46	6.28	19.3		51.7
HbA_1c_ (mmol/mol)	61.03	11.02	31.04		95.63
HbA_1c_ (%)	7.73	1.01	4.99		10.90
Glomerular filtration rate (ml/min)	73.77	21.44	22.04		120.00
CSA vagus nerve (mm^2^)	2.49	0.88	1.0		5.0
SSR hand latency (ms)	1.45	0.35	0.43		2.69
SSR hand amplitude (mV)	1764.43	1570.61	115.0		9415.0
SSR foot latency (ms)	1.89	0.44	0.85		2.94
SSR foot amplitude (mV)	747.07	873.42	0		5265.0
CIMT (mm)	0.97	0.24	0.6		2.0
NSS (0–10 points)	3.83	3.15	0		9
NDS (0–10 points)	6.60	2.61	0		10

Abbreviations: CIMT, carotid intima–media thickness; CSA, cross-sectional area; SSR, sympathetic skin response; NCS, nerve conduction study; NDS, Neuropathy Disability Score; NSS, Neuropathy Symptom Score.

**Table 2 medicina-59-00525-t002:** Pearson correlation.

Variable		SRR Hand Latency	SRR Hand Amplitude	SRR Foot Latency	SRR Foot Amplitude	CIMT	Age	Duration of Diabetes	BMI	HBA1c	GFR
CSA vagus nerve (mm^2^)	correlation	0.035	0.018	0.037	−0.161	0.097	0.060	0.163	0.155	0.062	−0.175
	*p*	0.715	0.847	0.740	0.092	0.314	0.530	0.091	0.110	0.524	0.073
SRR hand latency (ms)	correlation		−0.171	−0.050	0.173	0.048	0.190 *	0.175	−0.096	−0.023	0.057
	*p*		0.073	0.650	0.069	0.618	0.046	0.069	0.325	0.817	0.561
SRR hand amplitude (mV)	correlation			0.147	0.223 *	0.049	−0.160	−0.147	0.212 *	−0.024	0.038
	*p*			0.178	0.018	0.611	0.096	0.129	0.028	0.807	0.701
SRR foot latency (ms)	correlation				0.044	0.075	0.156	0.073	0.076	0.119	−0.102
	*p*				0.690	0.497	0.155	0.507	0.493	0.285	0.363
SRR foot amplitude (mV)	correlation					−0.173	0.036	−0.078	−0.017	−0.137	0.033
	*p*					0.072	0.706	0.421	0.862	0.160	0.740
CIMT (mm)	correlation						0.224 *	0.033	−0.058	0.141	−0.118
	*p*						0.020	0.738	0.559	0.151	0.234
Age (years)	correlation							0.440 **	−0.115	0.204 *	−0.636 **
	*p*							0.000	0.242	0.035	0.000
Duration of diabetes (years)	correlation								−0.104	0.122	−0.432 **
	*p*								0.285	0.212	0.000
Body mass index (kg/m^2^)	correlation									0.180	−0.153
	*p*									0.064	0.120
HbA_1c_ (mmol/mol)	correlation										−0.163
	*p*										0.096

* The correlation is significant at a level of 0.05. ** The correlation is significant at a level of 0.01. Abbreviations: BMI, body mass index; CIMT, carotid intima–media thickness; CSA, cross-sectional area; GFR, glomerular filtration rate; SSR, sympathetic skin response.

**Table 3 medicina-59-00525-t003:** Baseline characteristics of the three groups.

Group	N	Age Years (SD)	Gender Female:Male	Vagus Nerve CSA mm^2^ (SD)
Diabetes mellitus	111	66.3 (SD 10.05)	43:68	2.49 (SD 0.88)
Healthy controls	104	52.49 (SD 17.89)	46:58	2.26 (SD 0.59)
CIDP	41	65.44 (SD 12.91)	9:32	2.61 (SD 1.11)

Abbreviations: CIDP, chronic inflammatory demyelinating polyneuropathy; CSA, cross-sectional area; SD, standard deviation.

## Data Availability

The data used to support the findings of this study are available from the corresponding author upon request.

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
