# Peer review of "Evaluating Diagnostic Ultrasound of the Vagus Nerve as a Surrogate Marker for Autonomic Neuropathy in Diabetic Patients"

_medicina, 2023, doi:10.3390/medicina59030525_

Round 1
Reviewer 1 Report
The manuscript is very interesting, methodology is correct, and the main strength is the sample size of included patients; it also add new information to the usefulness of vagus nerve US for diabetic neuropathy.
However some major issues emerged after reading the manuscript.
MAJOR ISSUES
- Was a nerve conduction study performed? It would consolidate the clinical suspicion of diabetic neuropathy.
- Line 86 and 90: why two probes were used? It should be outlined in the limitations of the study.
- Line 118-121: add a reference for the standard methods used.
- Section 2.6: the results would be more robust if a regression analysis is performed. A correlation analysis is sometimes considered just a preliminary approach. I suggest to perform a regression analysis using CSA as dependent variables, and the other covariates collected and independent variables. If no significant beta-coefficient is seen, it would confirm your hypothesis.
- Table 1: simplify cells, for example using a single line with “yes/no”, instead of two lines for each cell.
- Line 153-156: add p-values in the main text.
- Table 2: provide the table as a triangular matrix, since there is a repetition of values, as known.
Author Response
1.1 Was a nerve conduction study performed? It would consolidate the clinical suspicion of diabetic neuropathy.
Reply: Yes. We included the information about NCS measurements into the text. Supplementary Figure S1 was introduced to depict the results based on diagnosis of polyneuropathy based on NCS. However, even for NCS based diagnosis of polyneuropathy no statistically significant difference of vagus nerve CSA could be found between patients with and without polyneuropathy. The results are reprted now in the text (lines 134-138, Table 1, lines 175-178).
1.2 Line 86 and 90: why two probes were used? It should be outlined in the limitations of the study.
Reply: The data of the different patient groups (diabetes patients, normal controls, and CIDP patients were derived from different studies where different probes were used. We included this in the part of limitations now. (line 303)
1.3 Line 118-121: add a reference for the standard methods used.
Reply: Done (line 129). We added references 11 and 12.
1.4 Section 2.6: the results would be more robust if a regression analysis is performed. A correlation analysis is sometimes considered just a preliminary approach. I suggest to perform a regression analysis using CSA as dependent variables, and the other covariates collected and independent variables. If no significant beta-coefficient is seen, it would confirm your hypothesis.
Reply: Yes, we did that and no significant beta-coefficient was found accordingly. We included the regression analysis as Supplementary Table S1 and included it into results and discussion (line 189-191, lines 280-284).
1.5 Table 1: simplify cells, for example using a single line with “yes/no”, instead of two lines for each cell.
Reply: Done (Table 1)
1.6 Line 153-156: add p-values in the main text.
Reply: Done (lines 171-174)
1.7 Table 2: provide the table as a triangular matrix, since there is a repetition of values, as known.
Reply: Done (Table 2)
Reviewer 2 Report
This review article contains updated topics in the etiology of diabetic autonomic neuropathy with special reference to the implication of vagus nerve. The results of this study provide important insights into current knowledge on the pathophysiology of this disease and contribute to the development of targeted therapy directed against damge vagus nerve. Considering the prevalence of diabetic autonomic neuropathy, this manuscript will attract broad range of readers. I do not have any critical comments.
1. The legend of the figure could be expanded to highlight specific elements you want the readers to pay attention to.
2. I propose to convert the results in the Table 2 into graphs.
3. Whether the authors have the data in Figures 2 and 3 separately for female and male? I propose to show the gender differences.
Author Response
2.1 The legend of the figure could be expanded to highlight specific elements you want the readers to pay attention to.
Reply: Done (Figure 1 lines 181-182, Figure 2 lines 212-215)
2.2 I propose to convert the results in the Table 2 into graphs.
Reply: We included selected scatterplots into the supplementary material. See line, Supplementary Figures S3 and S4. See also lines 184-187.
2.3 Whether the authors have the data in Figures 2 and 3 separately for female and male? I propose to show the gender differences.
Reply: We present Figure 2 and 3 separated according to gender in Supplementary Figure S2 and S5. See also lines 174-175 and lines 207-209.
Reviewer 3 Report
Dear authors,
I want to congratulate you on the chosen topic. Diabetes has long surpassed the status of a chronic disease, becoming a public health problem. Unfortunately, the disease starts at younger and younger ages, and I, as a pediatric nephrologist, can recognize that degenerative complications, once the prerogative of the third age, appear from young people. Indeed, the ultrasonographic method of measuring the vagus nerve is laborious and requires an experienced sonographer, but it is also non-invasive, which can be an advantage.
Returning to the work. I appreciated the presentation of the work method and the results. I would suggest developing the introduction, which, from my point of view, is too brief. Likewise, the discussion chapter could be improved, possibly by addressing other elements of autonomic neuropathy in diabetics, such as gastrointestinal, genitourinary and ocular, affecting thermoregulation and blood pressure; constipation, with urination problems; sexual dysfunction.
Anyway, the work is valuable for the doctors who care for the diabetic patient and may open a way to investigate this important complication, which is autonomic neuropathy. Besides, I saw that you have concerns in the field, materialized also through other articles with the same topic.
I wish you success and assure you of my support in publishing this article.
Author Response
3.1 I would suggest developing the introduction, which, from my point of view, is too brief.
Reply: We expanded the introduction accordingly (lines 50-61).
3.2 Likewise, the discussion chapter could be improved, possibly by addressing other elements of autonomic neuropathy in diabetics, such as gastrointestinal, genitourinary and ocular, affecting thermoregulation and blood pressure; constipation, with urination problems; sexual dysfunction.
Reply: We included this in the discussion (lines 290-294).
Round 2
Reviewer 1 Report
Authors answered to all my previous comments, and modified accordingly the manuscript, which has improved. I have no further comments.
Reviewer 3 Report
Dear authors, I want to thank you for the compliance and collaboration to make this article better. Obviously the introduction is better. The addition of supplementary material brought more originality to the text, making it easier to understand for other specialists interested in the topic, not only for neurologists and diabetologists. Overall, the work is clearly improved. Perhaps the discussions should have been developed and new references added. But I think that other articles in the field will follow, since you have shown a real interest in the subject. I congratulate you and I think that the article can be published in its current form.